# Probing muon $g - 2$ at a future muon collider

Jason Arakawa[1,2]⋆, Arvind Rajaraman[2]†, Taotao Sui[2,3]‡ and Tim M.P. Tait[2]◦

**1** Department of Physics and Astronomy, University of Delaware,
Newark, Delaware 19716, USA
**2** Department of Physics and Astronomy, University of California, Irvine, CA 92697-4575 USA
**3** Institute of Theoretical Physics and Research Center of Gravitation, Lanzhou University,
Lanzhou 730000, China

⋆ arakawaj@udel.edu , † arajaram@uci.edu , ‡ suitt14@lzu.edu.cn , ◦ ttait@uci.edu

## Abstract

The $4.2\sigma$ discrepancy in the $(g - 2)$ of the muon provides a hint that may indicate that physics beyond the standard model is at play. A multi-TeV scale muon collider provides a natural testing ground for this physics. In this paper, we discuss the potential to probe the BSM parameter space that is consistent with solving the $(g - 2)_\mu$ discrepancy in the language of the SMEFT, utilizing the statistical power provided by fitting event rates collected running at multiple energies. Our results indicate the importance of including interference between the BSM and the SM amplitudes, and illustrates how a muon collider running at a handful of lower energies and with less total collected luminosity can better significantly constrain the space of relevant SMEFT coefficients than would be possible for a single high energy run.

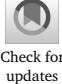
# 1 Introduction

The muon's anomalous magnetic dipole moment represents an enduring and statistically significant hint for physics beyond the Standard Model (BSM). The discrepancy in $a_\mu = (g-2)_\mu/2$ measured at BNL [1] has been solidified by the recent Fermilab results [2], shifting the central value slightly, and resulting in a higher combined significance of $4.2\sigma$. [3–22] At present, the combined result is:

$$\Delta a_\mu = a_\mu(\text{Exp}) - a_\mu(\text{SM}) = (251 \pm 59) \times 10^{-11}. \tag{1}$$

Potential BSM explanations span the gamut from light weakly coupled to heavy strongly coupled new particles. In specific realizations, the shift of the $g-2$ can be calculated from the additional vertex corrections that appear within the model at hand. On the other hand, if the new physics is heavy compared to the energy scales of interest, the framework of effective field theory (EFT) provides a robust set of tools that describes the low energy manifestation of UV physics in a more model independent manner. In particular, the Standard Model effective field theory (SMEFT) outlines the set of operators that are allowed by the symmetries of the SM [23–26].

If this discrepancy persists and increases as Fermilab collects more data and theoretical estimates for the SM prediction become more precise, it will cement $g-2$ of the muon as genuine BSM physics and it will become important to search for complimentary effects at other experiments in order to better understand its origin. A promising place to search for these effects would be at a future muon collider (MC). Key elements of the physics case to pursue a future high energy MC have been discussed extensively in the literature (e.g. Ref [27–31, 31–37] ), and the ability for a MC to probe the new physics responsible for the current measurements of the muon $g-2$ has been explored in Refs. [38–44], each of which argue that an $\mathcal{O}(10\,\text{TeV})$ scale MC could provide important information. Additionally, searches for leptonic magnetic dipole moments at colliders have been considered [45, 46].

In this article, we explore the processes $\mu^-\mu^+ \to \gamma h$ and $\mu^-\mu^+ \to Zh$, at a future high energy muon collider as means to learn about the underlying physics responsible for $g-2$ of the muon in the context where it is generated by heavy physics that is described at low energies by operators contained in the SMEFT. We pursue an approach that leverages fits of data collected at multiple center-of-mass energies to increase sensitivity. Instead of maximizing luminosity collected at a single collider energy, yielding an isolated event rate defined by a single cross section, we propose the use of multiple runs at different energies to understand the energy dependence of contributing processes. That is, we simulate data and fit the data to SM and SM+BSM expectations in an attempt to see small differences in the predicted rates. We examine the new physics (NP) contribution to the processes $\mu^-\mu^+ \to \gamma h$, $Zh$. Generically in the context of an EFT, the energy scaling of the SM and NP are drastically different, which provides the opportunity to leverage the fit to extract a relatively small component (with a different energy scaling) from the irreducible SM background.

# 2 Effective field theory description

Effects from new physics in the UV can be encoded in the IR via an effective field theory, consisting of the SM itself plus a tower of non-renormalizable terms encoding the residual effects of heavy states. The resulting theory can be organized in terms of the dimensionality of the BSM operators, where at energies far below the masses of fields which have been integrated out, the influence of the lower dimensional operators are expected to dominate over those with higher dimension. Demanding that the higher dimensional operators (linearly) re-

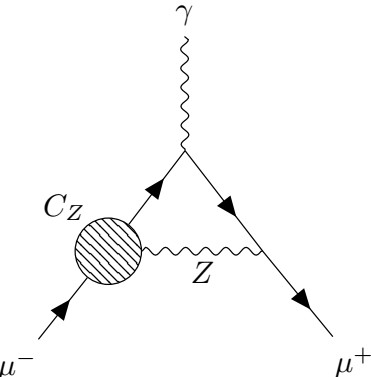

Figure 1: Representative diagram illustrating how $C_Z$ contributes to muon $g - 2$ at one loop.

alize the $SU(3)_c \times SU(2)_L \times U(1)_Y$ gauge symmetry structure of the SM yields the Standard Model Effective Field Theory which provides a robust framework describing the impact of new physics on SM processes. We consider the subset of operators within the SMEFT that provide contributions to the muon anomalous magnetic dipole moment. The lowest such operators are dimension-6, and are characterized by an energy scale $\Lambda$ and Wilson coefficients $C_B$, $C_W$:

$$\frac{C_B}{\Lambda^2}(H\overline{L}_2)\sigma^{\mu\nu}\mu_R B_{\mu\nu} + \frac{C_W}{\Lambda^2}(H\overline{L}_2)\sigma^{\mu\nu}\mu_R \tau^a W^a_{\mu\nu} + \text{h.c.} \tag{2}$$

where $H$ is the SM Higgs doublet, $L_2$ is the second generation lepton doublet, and $B_{\mu\nu}$ ($W_{\mu\nu}$) is the $U(1)_Y$ ($SU(2)_L$) field strength. After electroweak symmetry-breaking, these interactions mix into a modification of the muon's anomalous magnetic dipole moment:

$$\frac{C_\gamma}{\Lambda^2}(H\overline{L}_2)\sigma^{\mu\nu}\mu_R F_{\mu\nu} \rightarrow \frac{v\, C_\gamma}{\Lambda^2}\overline{\mu}_L \sigma^{\mu\nu}\mu_R F_{\mu\nu}\,. \tag{3}$$

where $v = 246\,\text{GeV}$ is the Higgs vev and $F_{\mu\nu}$ is the electromagnetic field strength. There is an analogous interaction with the $Z$ boson with coefficient $C_Z$, and related CP-violating terms if $C_B/C_W$ are complex-valued. The coefficients in the gauge and mass bases are related by the weak mixing angle: $C_\gamma = c_W C_B - s_W C_W$ and $C_Z = -s_W C_B - c_W C_W$, with $c_W \equiv \cos\theta_W$ and $s_W \equiv \sin\theta_W$.

$C_\gamma$ contributes to $(g-2)_\mu$ directly at tree level, whereas $C_Z$ contributes at one loop [41] (see figure 1):

$$\Delta a_\mu \sim \frac{\alpha}{2\pi}\frac{v\, m_\mu}{\Lambda^2}\left(C_\gamma - \frac{3\alpha}{2\pi}\frac{c_W^2 - s_W^2}{s_W c_W}C_Z \log\frac{\Lambda}{m_Z}\right). \tag{4}$$

(and additional one loop contributions from four-fermion interactions involving heavy quarks not considered here) [41], Since any physics at scales $\gg v$ must be approximately $SU(2)_L \times U(1)_Y$ invariant (and thus most naturally described by $C_{B,W}$), barring strangely tuned cancellations, it is likely that $C_\gamma$ and $C_Z$ will end up being similar in magnitude, and thus we expect that the contributions to $\Delta a_\mu$ will be typically dominated by $C_\gamma$. For the remainder of our discussion, we fix $\Lambda = 250\,\text{TeV}$, for which $C_\gamma$'s of order positive unity are required to explain the observed $\Delta a_\mu$.

# 3  Collider simulation

We simulate the processes $\mu^+\mu^- \rightarrow \gamma h$ and $\mu^+\mu^- \rightarrow Zh$ at tree level using `Madgraph5_aMC@NLO` [47], for different values of the muon collider beam energy. At leading order, there are both SM and SMEFT contributions, as shown in Figure 2, where the insertion of the SMEFT dimension-6 $C_\gamma$ operator is represented by the blob. Analytically, the cross section for $\mu^+\mu^- \rightarrow \gamma h$ is

$$\frac{d\sigma}{d\cos\theta} = \frac{e^2 y_\mu^2}{8\pi E^2}\frac{1}{1-\cos^2\theta} - \frac{e y_\mu C_\gamma}{32\pi\sqrt{2}\Lambda^2}(1+5\cos\theta) + \frac{C_\gamma^2 E^2}{32\pi\Lambda^4}(1-\cos^2\theta),  \quad (5)$$

where $y_\mu$ is the muon Yukawa coupling, $E$ is the energy of the muon beams, $\theta$ is the angle between the final state photon and the beam axis, and the mass of the muon has been neglected. The purely SM terms fall as $1/E^2$, as expected for renormalizable interactions. The BSM amplitude arises from a dimension six operator and thus grows with energy, resulting in the SM-BSM interference term being $E$-independent, and the pure BSM term growing as $E^2$.

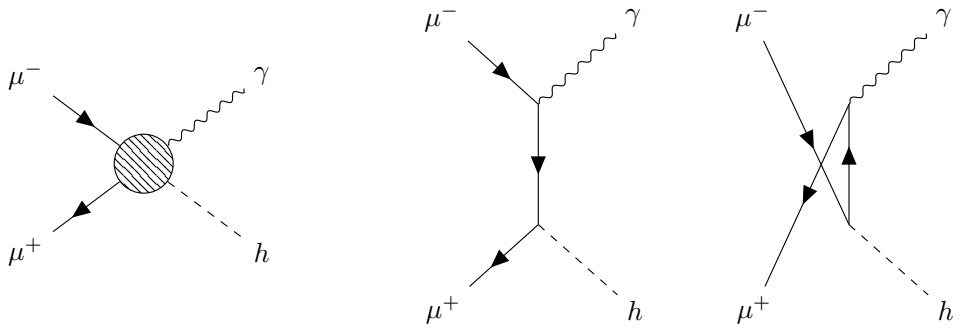

Figure 2: Contributions to $\mu^+\mu^- \rightarrow \gamma h$ from SMEFT operators (left) and the Standard Model (right).

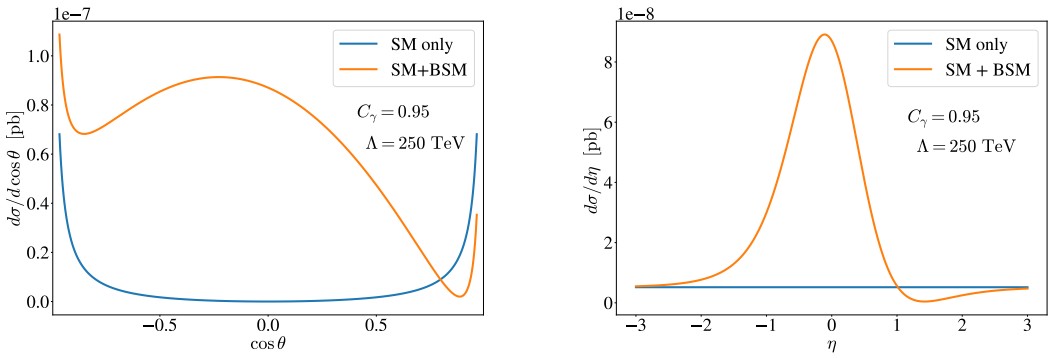

Figure 3: Differential cross section $d\sigma/d\cos\theta$ (left) and $d\sigma/d\eta$ (right) as a function of $\cos\theta$ and pseudo-rapidity $\eta$, respectively. Displayed are the purely SM case (blue), and the full SM+BSM case for $C_\gamma = 0.95$ (orange).

In addition, the SM contribution is sharply peaked toward forward region, because of the collinear singularity. In Fig. 3, we show the different cross sections for scattering angle $\theta$ and pseudorapidity $\eta \equiv -\ln\tan(\theta/2)$, for both $C_\gamma = 0$ (SM-only) as well as the case with $C_\gamma = 0.95$. These distributions are asymmetric in $\theta$ and $\eta$ because the interference term is

sensitive to the sign of $C_\gamma$; cases where these features are measurable thus allow the opportunity to reconstruct its sign, which is crucial in order to connect it to the observed deviation in $a_\mu$.

Given the high precision detectors envisioned for a future muon collider [48], we assume that the SM Higgs and/or $Z$ can be close to perfectly reconstructed, regardless of their specific decay channels, and that the uncertainties on the reconstructed final state energies and directions will be sufficiently small as to allow one to efficiently reject fake backgrounds without significant loss of signal events. To check this assumption, we calculate the contamination of the signal where a $Z$ boson in a $\gamma Z$ final state may be misidentified as a Higgs for different energy resolutions. In Table 2, assuming three benchmark choices of experimental resolution of 1%, 5% and 10%, we calculate the ratio of number of events that would be misidentified over the number of signal events. Below an energy resolution of $\sim 5\%$, the number of misidentified events is negligible. In light of the SM contribution's strongly peaked distribution in the forward directions compared to the BSM contributions tendency to populate the central region, we place a cut on the pseudo-rapidity, $|\eta_\gamma| < 1$ to enhance the significance of the BSM contributions, with a very modest loss of BSM signal.

## 4 Analysis

| $E$ | $\mathcal{L}$ |
|---|---|
| 1 TeV | 5 ab$^{-1}$ |
| 17 TeV | 15 ab$^{-1}$ |
| 18 TeV | 20 ab$^{-1}$ |

Table 1: Collider energies and corresponding integrated luminosities considered in the analysis.

| Exp. Resolution | $E_1, \mathcal{L}_1$ | $E_2, \mathcal{L}_2$ | $E_3, \mathcal{L}_3$ |
|---|---|---|---|
| 1% | 0 | 0 | 0 |
| 5% | $1.60 \times 10^{-5}$ | $3.82 \times 10^{-7}$ | $3.069 \times 10^{-7}$ |
| 10% | 779.71 | 18.65 | 14.99 |

Table 2: Ratio of the number of events $N_{Z\gamma}/N_{h\gamma}$ for different choices of experimental resolution and for each of the energies and luminosities considered in Table 1.

For the $\mu^+\mu^- \to \gamma h$ channel, the number of events at luminosities envisioned at a future high energy muon collider is $\mathcal{O}(1)$, requiring Poisson statistics in order to evaluate the uncertainties and statistical significance of an observed deviation. We generate simulated datasets corresponding to the expected event rates (subject to Poisson fluctuations) of both the SM ($C_\gamma = 0$) and BSM+SM with various $C_\gamma \neq 0$, for a set of proposed collider energies and luminosities shown in Table 1. This set of energies and luminosities could correspond to a future muon collider which runs initially at lower energies to make precision measurements of the properties of the Higgs boson and top quark, and then accumulates additional data at a couple of very high energies. We show the predicted event rate as a function of energy and the differential event rate as a function of $\eta$ in Fig. 4 (left and right, respectively). It would be interesting, though beyond the scope of this work, to explore how these choices could be optimized.

We fit the energy dependence of the BSM+SM cross section to both data sets, extracting $C_\gamma$ for each realization. Sampling a large number of pseudo-datasets maps out the distributions

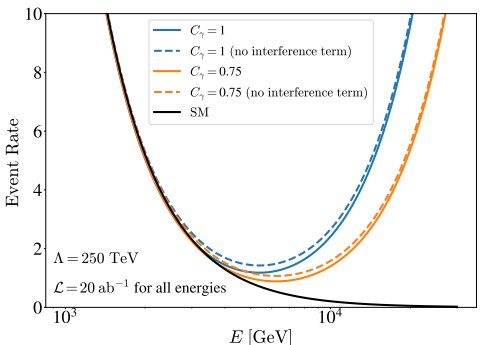
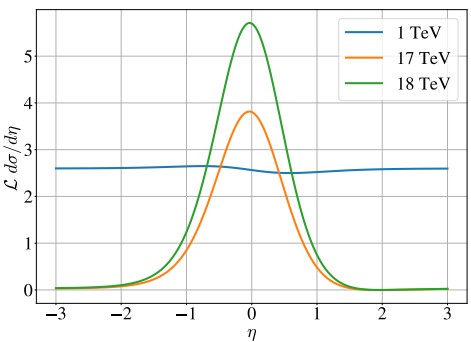

Figure 4: Left: Event rate as a function of energy for various choices of $C_\gamma$. The dashed lines represent the predicted event rate without the interference term. We use a constant luminosity of $\mathcal{L} = 20$ ab$^{-1}$ for all energies for demonstrative purposes. Right: The differential event rate as a function of $\eta$ for each of the energy and luminosity benchmarks listed in Table 1. The interference term manifests itself in generating an asymmetry about $\eta = 0$.

of the extracted $C_\gamma$, as displayed in Fig. 5 for the specific choice of $C_\gamma = 0.65$. The distribution of the ensemble of BSM+SM simulations provides a determination of the expected spread in the inferred $C_\gamma$ (around our assumed $C_\gamma = 0.65$) due to the expected statistical fluctuations, whereas the distribution of the extracted $C_\gamma$ in the SM-only ensemble (which is highly pixelated because event rates in any particular realization are integers) reveals the value of $C_\gamma$ that can typically be excluded if there is no BSM contribution. The overlap of the two distributions determines the confidence level with which $C_\gamma$ can be probed. We find that

$$-0.63 \lesssim C_\gamma \lesssim 0.65 \qquad (\Lambda = 250 \text{ TeV}), \tag{6}$$

would remain viable at the 95% confidence limit (CL), whereas $C_\gamma \gtrsim 0.85$ would lead to evidence for new physics at greater than $3\sigma$.

The process $\mu^+\mu^- \to Zh$ is also sensitive to both $C_B$ and $C_W$, in the orthogonal combination that is $C_Z$. The SM rate for this process is much larger than for the $\gamma h$ final state, $\sigma_{Zh,\text{SM}} \approx 122 \text{ ab} \left(\frac{10 \text{ TeV}}{\sqrt{s}}\right)^2$ due to the additional $s$-channel diagram. While this implies a larger irreducible background, it also enhances the interference terms between SM and BSM contributions. Consequently, the expected number of events is typically large enough that Gaussian statistical analysis is sufficient to extract the expected reach in $C_Z$, and would restrict

$$-8.6 \lesssim C_Z \lesssim 8.0 \qquad (\Lambda = 250 \text{ TeV}), \tag{7}$$

at 95% CL.

Putting these together, we find that a muon collider running in this configuration of energies and luminosities can probe a large swath of the region of the SMEFT able to explain the observed measurement of $\Delta a_\mu$. In the left panel of Figure 6, we show the bands of $C_\gamma$ and $C_Z$ consistent with the observed measurement at one and two $\sigma$. The portion shaded red would result in an observable deviation in the process $\mu^+\mu^- \to \gamma h$, whereas the portion shaded blue can be probed via $\mu^+\mu^- \to Zh$. A small wedge around $(C_\gamma, C_Z) \simeq (0.5, -5)$ would remain untested. Conversely, if no deviation were to be observed, nonzero $(C_\gamma, C_Z)$ would be ruled out at 95% CL. except for the violet shaded region around $(0,0)$, shrinking the viable parameter space capable of explaining $\Delta a_\mu$ to a large degree. In the right panel of Figure 6, we translate these regions into the SMEFT coefficients $C_B$ and $C_W$.

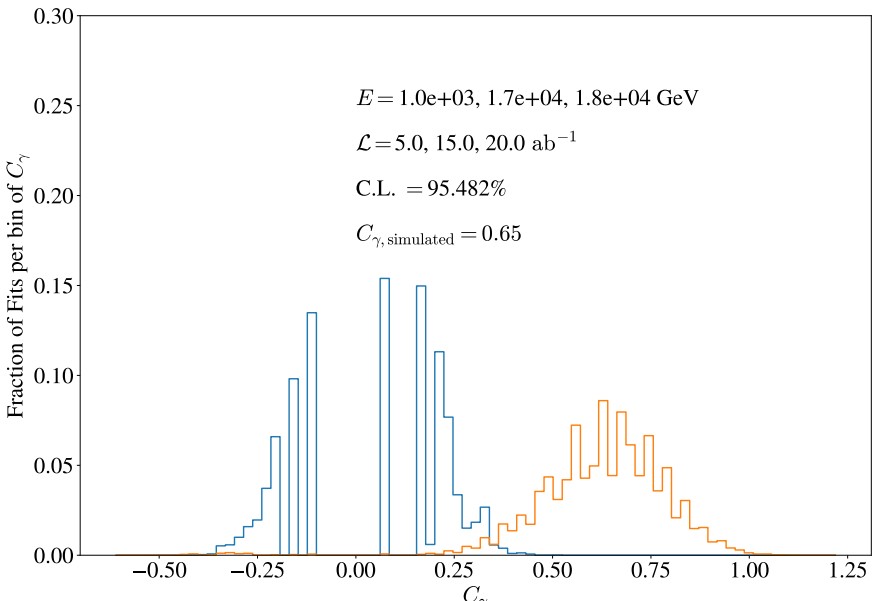

Figure 5: The distributions of fitted values for $C_\gamma$ based on generated pseudodata corresponding to SM-only (blue) and SM+BSM (orange) rates. The three energy and luminosity combinations used are displayed in Table 1.

# 5 Conclusions and Outlook

The discrepancy observed in the muon $g-2$ provides a long-standing hint towards BSM physics. A future high energy MC provides a unique opportunity to probe the potential BSM solutions. We find that a future MC can be sensitive to such BSM physics by utilizing the predicted energy scaling of the SM+BSM cross section. In particular, we illustrate this with a benchmark point, which defines the energies and luminosities of the multiple runs. Our benchmark considers energies of $E = 1\,\text{TeV}, 17\,\text{TeV}$, and $18\,\text{TeV}$, with integrated luminosities of $\mathcal{L} = 5\,\text{ab}^{-1}, 15\,\text{ab}^{-1}$, and $20\,\text{ab}^{-1}$ respectively. It would be interesting in the future to explore what configurations optimize the ability to probe $g-2$ based on minimal integrated luminosity data sets.

# Acknowledgements

**Funding information** This work was supported in part by the NSF via grant number PHY-1915005. The work of JA was supported in part by NSF QLCI Award OMA - 2016244.

SciPost Phys. **16**, 072 (2024)



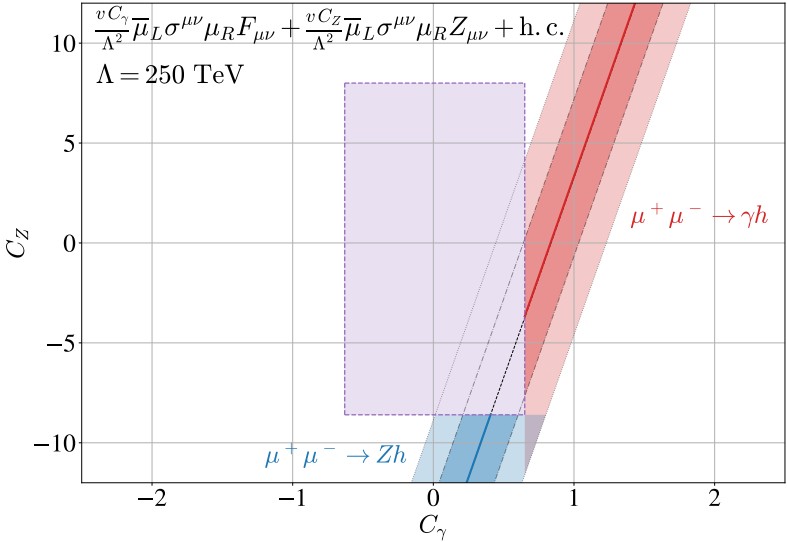

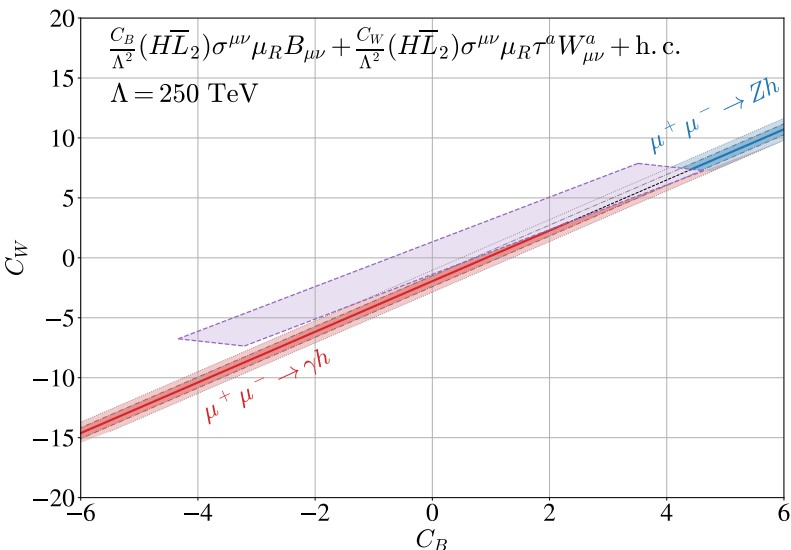

Figure 6: Parameter space that the muon collider can probe in the space of $C_\gamma$, $C_Z$ (top) and $C_B$, $C_W$ (bottom). The parameter space consistent with $\Delta a_\mu$ at one and two $\sigma$ are indicated by the bands, with the portion leading to an observable deviation in $\mu^+\mu^- \to \gamma h$ shaded red and the corresponding region for $\mu^+\mu^- \to Zh$ shaded blue. The violet region indicates the allowed parameter space at 95% CL if no deviation from the SM event rate is observed in the collider searches.

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
