# Peer review of "Probing Muon $g-2$ at a Future Muon Collider"

_SciPost Physics, doi:SciPost Phys. 16, 072 (2024)_

## Round 1 · Referee Report · Anonymous (Referee 1) · 2023-8-22

Strengths

1- Straight forward and easy to follow 2-The derived results possess the potential for application in forthcoming model-specific analyses.

Weaknesses

1-Scope Concern: The analysis seems restrictive, primarily focusing on a singular channel.
2-Depth Issue: SMEFT serves predominantly as a parameterization with no proposed underlying theories.

Report

The manuscript proposes an exploration of the g-2 muon discrepancy via the processes µµ → γh/Zh in a prospective muon collider. The manuscript first introduces relevant dimension-6 effective operators, then calculates the contribution to the muon's anomalous dipole moment at leading order. In section III the manuscript calculates the cross section for the process µµ → γh. In section IV, simulated datasets are used to put constraint on specified operators.

The idea presented in the paper is rather standard. Rather, such analysis is not done because we do not yet have such a detector in a near future. However as the reviewer aware, current analysis are only done for LHC processes.

The manuscript may be more appropriately suited for SciPost Core rather than SciPost Physics.

Requested changes

Some additional queries and suggestions:

1-The study appears limited to certain channels. Have the authors contemplated including others, such as µe -> µe scattering, to potentially constrain additional parameters?
2-Rather than implementing an energy cutoff, might it be more streamlined to set a boundary on C/Λ^2?
3-How many events were simulated?
4-In Figure 4, a clarification regarding the shape of the SM-only histogram would be beneficial.
5-These operators may already be constrained using existing LHC results, e.g. via h/Z -> µµ(γ). How do the presented result compare with these existing bounds?
6-A further enhancement would be a discussion on potential underlying BSM theories, even if speculative.

  • validity: good
  • significance: ok
  • originality: low
  • clarity: high
  • formatting: excellent
  • grammar: excellent

Author:  Jason Arakawa  on 2024-01-22  [id 4270]

(in reply to Report 1 on 2023-08-22)
Category:
remark
answer to question
pointer to related literature

We thank the referee for reviewing our manuscript. We address the comments below.

1- Our study is limited to a future muon collider, but this is an interesting question worthy of following up.

2- Although it's true that $C/\Lambda^2$ is what is being bounded, it's common convention in much of the SMEFT literature to parameterize it this way.

3- We simulated 50,000 events to generate the pseudo-data. We checked that our results converged, and didn’t change significantly by adding more points.

4- We explain in the text that the pixelation and shape comes from the number of events being small integers. For the SM only event rate, we determine what values of C with a fit to the SM+BSM cross section with Poissonian statistics. This means that since there’s some noise in the pseudo data coming from simulating statistics, the fit never finds C = 0.

5- Currently they are not sensitive, but the following reference proposes the analysis by the LHC for the related tau quantity: H→τ+τ−γ as a probe of the ττ magnetic dipole moment Iftah Galon, Arvind Rajaraman, Tim M. P. Tait (Published in: JHEP 12 (2016) 111 [1610.01601 [hep-ph]])

6- We thank the referee for bringing up the important issue of connecting to UV physics. This is beyond the scope of the paper, where we adopt the EFT framework to capture any possible effects by UV physics.

---

## Round 1 · Referee Report · Francesco Giovanni Celiberto (Referee 2) · 2023-8-27

Report

In this short research article the Authors highlight the discovery potential of a future multi-TeV muon colliding machine. In particular, evidence is provided that such a collider will serve as a useful tool to shed light on the 4.2 sigma discrepancy on the muon anomalous magnetic dipole moment.

From an operational perspective, the Authors rely upon the Standard-Model Effective Theory (SMEFT), namely one of the most powerful formalisms to explore the BSM space of parameters.
As reference processes, the Authors consider the γh and the Zh production in muon annihilations.

The core outcome of this study is the impact of SM+BSM interference terms, as well as the possibility of getting mora and more precise space-constraints of SMEFT coefficients via a muon collider running "a handful of lower energies and with less total collected luminosity", rather then what would be achieved via a single high-energy run.

I believe that the manuscript reaches the standards of quality, robustness, and novelty for SciPost Physics. The research design is appropriate and the methodology employed is adequately described. Overall, the manuscript also clearly written and suitably formatted.

Before to be considered for publication, however, I propose that the Authors take care of some minor aspects, reported below in order of appearance in the text.

Requested changes

  1. Fourth/fifth line of the introduction: "have been solidified" -> "has been solidified;

  2. Introduction, paragraph below Eq. (1): some relevant references to EFT in general and, more in particular, to SMEFT should be added;

  3. Towards the end of the introduction: the "NP" acronym needs to be spelt when introduced for the first time.

  • validity: high
  • significance: good
  • originality: ok
  • clarity: top
  • formatting: good
  • grammar: excellent

Author:  Jason Arakawa  on 2024-01-22  [id 4271]

(in reply to Report 2 by Francesco Giovanni Celiberto on 2023-08-27)

We thank the referee for the careful review of our manuscript. We have made the changes requested.

---

## Round 1 · Referee Report · Anonymous (Referee 3) · 2023-8-29

Strengths

  1. This analysis considers both the energy dependence of the BSM contributions to the muon g-2, and the interference with the SM amplitudes. This is a novel approach which has not been considered before.
  2. The paper is very clearly written and concise.

Weaknesses

  1. I find the assumption of zero background at the bottom of p. 2 extremely unrealistic. Even for a future collider, some nonzero fake rate as considered in Ref. [37] is surely an important aspect of the analysis.
  2. A number of additional plots could make the analysis easier to follow. For example, plots of the number of events in each cos theta bin for the three different energies considered would make immediately clear how the different energy scaling of the interference term interplays with the change in luminosity at the different center-of-mass energies.

Report

This paper is generally suitable for publication in SciPost Physics.

Requested changes

The authors should address each of the weaknesses noted, by: 1. showing how their results change as a function of the assumed rate of Zs faking Higgses, and 2. adding additional plots to show the energy dependence of the event rate

  • validity: high
  • significance: high
  • originality: high
  • clarity: high
  • formatting: excellent
  • grammar: excellent

Author:  Jason Arakawa  on 2024-01-22  [id 4272]

(in reply to Report 3 on 2023-08-29)
Category:
answer to question

We thank the referee for reviewing and thoughtfully suggesting additions to our paper. We address each of the comments below.

1- We have modeled the contamination of the signals to include what the background rate is depending on the energy resolution of the detector. We find that for energy resolutions below 5%, the background is negligible.

2- We’ve made a plot to show the energy and angular dependences of the event rate.

---

## Round 2 · Referee Report · Anonymous · 2024-2-12

Report

I appreciate the authors' careful considerations of my suggested edits. These changes are very well implemented and I am happy to recommend the paper for publication.

---

## Round 2 · Author Response

We thank the referees for the careful review of our manuscript and for providing thoughtful suggestions to improve our work. We have addressed each of the comments either in the manuscript or via a direct reply to the report.

---

## Round 2 · List of Changes

Added additional table and description of modeling contamination of the signal by background events.
Added two plots to demonstrate the energy and angular dependence of the event rate.
Added references throughout.
Fixed grammatical errors and typos.

---

## Editorial Decision

published